# The Role of the Gut Microbiome in Nonalcoholic Fatty Liver Disease

**DOI:** 10.3390/medsci6020047

**Published:** 2018-06-05

**Authors:** Sanjoy Roychowdhury, Praveen Conjeevaram Selvakumar, Gail A.M. Cresci

**Affiliations:** 1Department of Inflammation & Immunity, M17, Cleveland Clinic, 9500 Euclid Avenue, Cleveland, OH 44195, USA; roychos@ccf.org; 2Department of Pediatric Gastroenterology, M17, Cleveland Clinic, 9500 Euclid Avenue, Cleveland, OH 44195, USA; conjeep@ccf.org; 3Director for Nutrition Research Center for Human Nutrition, M17, Cleveland Clinic, 9500 Euclid Avenue, Cleveland, OH 44195, USA

**Keywords:** gut microbiome, NAFLD, probiotics, apoptosis, necroptosis

## Abstract

Nonalcoholic fatty liver disease (NAFLD) is the leading cause of chronic liver disease, with prevalence increasing in parallel with the rising incidence in obesity. Believed to be a “multiple-hit” disease, several factors contribute to NAFLD initiation and progression. Of these, the gut microbiome is gaining interest as a significant factor in NAFLD prevalence. In this paper, we provide an in-depth review of the progression of NAFLD, discussing the mechanistic modes of hepatocyte injury and the potential role for manipulation of the gut microbiome as a therapeutic strategy in the prevention and treatment of NAFLD.

## 1. Introduction

Nonalcoholic fatty liver disease (NAFLD) is defined as the excessive accumulation of triglycerides in ≥5% of hepatocytes in the absence of significant alcohol consumption (less than 20 and 30 g per day in women and men, respectively), or viral, autoimmune, metabolic or drug-induced liver disease [1,2,3]. Recent evidence show that NAFLD could be a precursor to the development of progressive metabolic disease, rather than just an association with metabolic syndrome [4]. NAFLD has been shown to be a risk factor for the development of metabolic syndrome, type 2 diabetes, and cardiovascular diseases [5]. NAFLD is a dynamic disease encompassing a broad clinicopathologic spectrum, starting with isolated hepatic steatosis (simple steatosis), then progressing to varying degrees of necroinflammation, and subsequently leading to fibrosis and eventually cirrhosis [6,7]. Although NAFLD increases liver-related morbidity and mortality, cardiovascular disease and extrahepatic malignancy are the most common causes of death amongst patients with NAFLD [8]. In addition to liver-related morbidity and mortality, NAFLD may also be associated with chronic kidney disease, hypothyroidism, polycystic ovarian syndrome, obstructive sleep apnea (OSA), osteoporosis, and colorectal cancer [9,10,11,12]. 

Obesity has been shown to affect almost every organ system, and is associated with cardiovascular diseases, type 2 diabetes, hypertension, and even certain cancers, causing exponential increases in healthcare costs [13,14]. Increased consumption of energy dense foods and decreased physical activity are the most important culprits implicated in the dramatic increase in obesity prevalence globally. The impact of this obesogenic environment on obesity and its related comorbidities is further intensified by our understanding of the increased longevity and decreased occurrence of chronic diseases in the Okinawan population, related to their favorable nutritional profile and other lifestyle habits, compared to those of the rest of the world [15]. Despite this formidable contribution from energy dense diets, other factors are also believed to contribute to obesity. There has been a growing body of evidence surrounding the role of the gut microbiome in the pathogenesis of obesity [16]. Similarly, as expected, the gut microbiome has also been linked with chronic liver diseases, including NAFLD. With trillions of microorganisms, the gut microbiome is currently considered an important organ in the body which interacts with other organ systems in a complex way through its specific metabolites, hormones, and neurotransmitters [17]. In this review, we aim to study the recent data on the gut-liver axis, its role in the development and progression of NAFLD, and nutritional approaches which may modify the gut microbiome as a strategy to mitigate NAFLD.

## 2. NAFLD/NASH Epidemiology

Ludwig officially reported nonalcoholic steatohepatitis (NASH) associated with fibrosis and cirrhosis in liver biopsies of patients with obesity, diabetes, and cholelithiasis [18]. This was quickly followed by the first reports of NASH in obese children with nonspecific abdominal pain and abnormal liver function tests [19]. Since the first report in 1980s in both adults and children, our understanding in the pathophysiology, prognosis and burden of nonalcoholic fatty liver disease (NAFLD) on the health care system has greatly improved.

Keeping up with the rising obesity epidemic, NAFLD has now become the leading cause of chronic liver disease amongst all ages in Westernized countries, and has become the second most common cause of liver transplantation among adults [20,21]. It is noteworthy that NAFLD prevalence varies widely, depending on the study population and diagnostic method used. NAFLD is believed to affect approximately one-third of the population in the United States, with an estimated prevalence of 27–34% [22]. From a global perspective, the prevalence of NAFLD is approximately between 20 and 30% in the Western world, and 5 to18% in Asian countries [23,24]. In South America, prevalence is estimated to be between 23 and 30% [25]. Among European countries, the prevalence of NAFLD varies widely, from 5–44%, depending on the diagnostic modality used, with an average of 23.7% [26]. In developing nations, NAFLD prevalence has an interesting paradigm: higher prevalence rates are seen in urban populations compared to rural areas, likely due to exposure to traditional diets and lifestyle in rural areas, as opposed to the Westernized diet and lifestyle in urban populations [27,28]. NAFLD is shown to be relatively less prevalent in African countries [25]. In general, countries with higher economic status are reported to have higher prevalence rates [29]. More importantly, populations with other co-morbidities, such as diabetes and morbid obesity, are likely to have higher prevalence of NAFLD, with 70% and 90% NAFLD prevalence rates, respectively [30,31]. Ethnicity and race also factor into NAFLD prevalence, with Hispanic Americans having the highest prevalence, and African Americans the lowest [32]. The global prevalence of NASH is estimated to be between 2–3% among the adult population [33]. Amongst children, prevalence studies are more limited. Currently, The Study of Child and Adolescent Liver Epidemiology (SCALE) is the only study for children; this study is based on liver histology. After reviewing the clinical records and liver histology of 742 children (aged 2–19 years) who died from unnatural causes, this study reported 38% NAFLD prevalence among obese children. Moreover, the estimated prevalence of NAFLD in the overall teenager population was 17%, as opposed to 0.7% in children between 2 and 4 years of age, suggesting that pediatric NAFLD rates increase with age [20]. Using elevated plasma alanine aminotransferase (ALT) as a surrogate NAFLD marker, National Health and Nutrition Examination Survey (NHANES) reports an estimated NAFLD prevalence amongst US adolescents of 8–10.7%, with an approximately two-fold increase in the past two decades [34,35]. In contrast, estimated prevalence was shown to be only 2.5% using ultrasound to diagnose adolescent NAFLD [36].

## 3. Gut Microbiome

The human intestinal tract houses 10–100 trillion microbes, termed gut microbiota. Combined with their genetic material, these microbes comprise the gut microbiome (Figure 1) [37]. The gut microbiome is integral in maintaining intestinal homeostasis by aiding with nutrient digestion, metabolism, immune function, and barrier protection, and is therefore considered to be a functional organ [38]. Although gut microbiome is established by age 3, throughout life, multiple factors can alter its diversity, including medications (e.g., antibiotics), geography, stress, and diet. Using 16S rRNA sequence analysis, variance in principle coordinate spacing of bacterial community structures was identified in humans consuming different diets, such as plant-based versus meat-based [39]. The abundance of the two major microbial phyla, Firmicutes and Bacteroidetes, varies with body fat composition, with increased Firmicutes and deceased Bacteroidetes with greater adiposity [40]. This observation is found in mice and humans. Germ-free animals, which are characteristically lean despite consuming more food than conventional mice, increase in adiposity upon fecal material transfer from conventional mice, due to increased dietary energy extraction [41]. Interestingly, when obese humans changed their diet to restrict fat or carbohydrate, their gut microbiota shifted from an “obese” to a “lean” phenotype [40]. Increased Bacteroidetes and decreased Firmicutes numbers were found in obese children and adolescents with NASH, compared with normal weight patients [42]. In the obese group and the NASH group, depleted Firmicutes numbers were mostly accounted for by a decrease in families Lachanospiraceae and Ruminococcaceae. *Blautia* and *Faecalibacterium*, the most abundant genera in the Firmicutes phylum, showed the greatest reduction in abundance. Elevated blood alcohol levels in NASH patients correlated with elevated alcohol-producing bacteria [42].

Dietary changes also influence bacterial metabolism, which consequently alters the metabolic byproducts of the gut microbiota. The preferential food source for the gut microbiota is fermentable soluble fiber. As humans do not possess the enzymes to digest fibers, they escape digestion and are later digested and fermented by the gut microbiota, to yield short chain fatty acids [43]. Changes in microbiota metabolic byproducts can occur as quickly as within 1 day, with large shifts within 1 week of a dietary change [2]. 

## 4. Gut Dysbiosis and Bile Acids

Obesity is associated with the consumption of a Western Diet, (e.g., high fat, high sugar, low fiber). This diet is linked with reduced gut microbial diversity at the species and genus level, otherwise known as gut dysbiosis. Increased total bile acid levels are found in people with obesity, and in response to a meal with larger percentages of dietary fat [44].

Bile acids, detergents that promote solubilization, digestion, and absorption of dietary lipids, are synthesized from cholesterol in hepatocytes, conjugated by amino acids glycine or taurine, and carried in bile to the gallbladder. In response to a meal, and release of cholecystokinin from the duodenum, the gallbladder contracts and releases conjugated bile acids into the duodenum to aid in lipid absorption in the small intestine. Bile salts are reabsorbed through active transport in the distal ileum, complexed to plasma proteins, returned to the liver, and secreted back into bile [45]. During enterohepatic circulation, bile acids interact with gut microbiota in the small intestine, and undergo deconjugation, catalyzed by the enzyme bile salt hydrolase, and hydroxyl group oxidation. The bile salts that escape enterohepatic recirculation reach the colon and get converted into secondary and tertiary bile acid metabolites by intestinal bacterial transformations [45].

Bile acids are pleiotropic signaling mediators in metabolism and inflammation through interactions with bile acid receptors, the Farnesoid X receptor (FXR), and the G-protein coupled receptor TGR5 [45]. The gut microbiome has been shown to affect bile acid mechanisms not only by influencing the bile acid pool, but also the activation of bile acid receptors [46]. The intestinal immune functions and barrier are regulated by bile acid transformation and receptor activation; therefore, gut dysbiosis is associated with altered bile acid metabolism and inflammation [47]. 

Alterations in bile acids are associated with obesity, and NAFLD/NASH. Fasting and postprandial serum bile acid levels are elevated with NAFLD/NASH, and circulating bile acids correlate with NASH severity in both adult and children [44]. Advanced fibrosis and NASH is associated with higher levels of fecal conjugated primary bile acids [44]. There is an emerging consensus that the severity of NAFLD/NASH can be determined via the metagenomic signature of the gut microbiome [48].

## 5. NAFLD Pathophysiology

NAFLD is considered to be a multifactorial disease involving multiple intracellular signaling pathways. Obesity and insulin resistance are believed to initiate hepatic steatosis. Further inflammatory cascades lead to liver injury mediated by interactions between various organ systems, including adipose tissue and the gut microbiome [49]. Moreover, genetic polymorphisms, such as Patatin-like phospholipase domain-containing protein 3 (*PNPLA3*), transmembrane 6 superfamily member 2 gene (*TM6SF2*) and membrane bound O-acyltransferase domain containing 7 gene (*MBOAT7*), are also associated with hepatic steatosis and fibrosis [49].

The intestine and liver are tightly linked via the portal vein. Recently, there has been considerable interest in the role of the gut-liver axis in the development and progression of NAFLD. The possible role of the gut microbiota in chronic liver disease was first reported in 1921 by Hoefert. Since then, there has been increasing evidence regarding the role of the gut-liver axis dysfunction in the development and progression of NAFLD [50,51]. While a healthy gut microbiota composition produces beneficial byproducts, overgrowth of pathogenic bacteria leads to the production of negative byproducts, such as endotoxin. Altered gut permeability, associated with the consumption of a Western diet and obesity, is linked with gut dysbiosis and negative microbial metabolites, which can reach the liver via portal circulation, and interact with toll-like receptors on hepatocytes to mediate this inflammatory cascade [52]. While obesity-driven intestinal permeability is clinically appreciated, the molecular mechanisms are not fully known. Recent research found high-fat diet (HFD)-induced gut dysbiosis in mice to be associated with impaired intestinal barrier function maintained by the mucin layer and anti-microbial peptides [53]. Of the tight-junctional proteins, organ and tissue-specific claudin proteins and subsequent signaling pathways were modulated by HFD-induced obesity in mice, with obesity-associated secretomes contributing to these changes [54].

## 6. Adipose Tissue Dysfunction and NAFLD

Adipose tissue dysfunction plays a critical role during NAFLD progression and related [55] metabolic syndrome. Inflammation of mesenteric visceral fat is implicated in the development of metabolic abnormalities associated with type 2 diabetes (T2DM) and NAFLD. HFD-induced adipose tissue dysfunction is associated with altered adipose tissue architecture, adipocyte apoptosis, and increased numbers of infiltrating macrophages, localized around dying adipocytes to form crown-like structures [56]. These changes in adipose architecture and related systemic consequences lead to altered secretion of adipose-derived cytokines, chemokines, and adipokines. Adipokines are the adipose tissue-derived hormones which regulate energy metabolism. Adiponectin, a potent adipokine, has been reported to reduce hyperlipidemia and insulin resistance. Low levels of adiponectin were detected in patients with NASH [57]. High-fat diet-induced steatohepatitis and fibrosis are exacerbated in adiponectin-deficient mice [58]. Collectively, these results indicate that high fat diet-induced adipose inflammation contributes to metabolic abnormalities during the development of NASH or T2DM, by modulating secretion of adipokines.

The link between an altered microbial signature in the intestine and adipose inflammation during the development of metabolic syndrome remains unclear. A recent study by Newdorp and colleagues detected the presence of *Ralstonia picketti*, a gram negative intestinal bacteria, in visceral adipose tissue from T2DM patients [59]. Glucose tolerance is markedly reduced in diet-induced obese mice exposed to *R. picketti*, compared to vehicle-exposed counterparts. Results from this study suggest that the translocation of intestinal bacteria to nearby fat depots triggers inflammation and glucose intolerance by modulating the secretion of adipokines, including adiponectin.

## 7. CYP2E1 and NAFLD

Obesity and the consumption of a high-fat containing diet disturb fatty acid metabolism, causing an accumulation of fatty acids and triglycerides in the liver. These lipids and their metabolites induce mitochondrial oxidative stress through upregulation of a microsomal enzyme, cytochrome P450 2E1 (CYP2E1). Although the predominant role of CYP2E1 is to metabolize drugs and xenobiotics, increased CYP2E1 activity results in excessive reactive oxygen species (ROS) generation, leading to oxidative stress [60]. CYP2E1 also metabolizes endogenous substrates including steroids and polyunsaturated fatty acids, including linoleic acid and arachidonic acid [61]. Superoxide is a member of the ROS family and a by-product of the CYP2E1 catalyzed-reaction, which utilizes molecular oxygen as its substrate. Increased activity of CYP2E1 is therefore associated with the excessive production of superoxide, which leads to generation of additional ROS molecules. Elevated expression of CYP2E1 has been detected in the livers of patients with NASH, as well as in rodents exposed to a fat-enriched diet. Mice deficient in CYP2E1 are protected from HFD-induced insulin resistance, oxidative stress and fibrosis, indicating a direct role of CYP2E1 in NASH and T2DM [62]. In addition to the liver, an elevated expression of CYP2E1 in adipose tissue has also implicated HFD-induced glucose intolerance through an altered translocation of the glucose transporter, GLUT4 [63]. Although, the role of CYP2E1 and the microsomal ethanol oxidizing system (MEOS) is well-established in alcoholic liver disease [64], the role of CYP2E1 in NAFLD is still not clear. Further studies are required to unravel the cell-specific roles of CYP2E1 in the setting of NAFLD or NASH.

## 8. Modes of Hepatic Cell Death in NAFLD

NAFLD progresses through multiple stages of hepato-pathologies, including liver steatosis, increased inflammation, and hepatocyte injury, and eventually the deposition of extracellular matrix proteins leading to liver fibrosis and cirrhosis [65]. Hepatocyte ballooning and death are implicated in NAFLD progression. During the progression of NAFLD, multiple programmed cell death (PCD) pathways are activated in the liver, including caspase-dependent apoptosis [66], pyroptosis [67], and caspase-independent necroptosis (Figure 2) [68]. Here, we will review the current literature regarding the differential contributions of multiple hepatocyte death modalities during the progression of NAFLD and NASH, and discuss future treatment opportunities.

Apoptosis, a caspase-driven PCD pathway, has been identified as a key player during the progression of NAFLD. This form of cell death is characterized by nuclear fragmentation, cell shrinkage, and blebs on the plasma membrane. Apoptosis follows either an intrinsic or extrinsic pathway. Intrinsic pathways are triggered by intracellular signals such as ROS, while extrinsic pathways are induced by external stimuli, including tumor necrosis factor α (TNFα), FasL, or TNFα-related apoptosis inducing ligand (TRAIL) circulating pro-death ligands. The protein Bid bridges between these two pathways. Plasma cytokeratin 18 (CK18), a by-product of hepatocyte-caspase activity, has been identified as a novel biomarker for liver fibrosis in both adult [66] and pediatric [69] NAFLD cases. Hepatocyte apoptosis can be triggered by a direct toxic effect of fatty acid intermediates, known as lipoapoptosis, or due to the increase of pro-death ligands, including TNFα, FasL or TRAIL, in circulation. Several pro-apoptotic proteins, including caspase 3 [70], Bim, and PUMA have been implicated in lipoapoptosis. Mice deficient in CD95 and TNFR1 are protected from HFD-induced apoptosis and obesity, indicating that apoptosis contributes to NAFLD progression. Although the majority of mouse studies utilize models with global deletion of pro-apoptotic proteins, a few recent studies using tissue-specific knockout mouse models have also confirmed that hepatocyte-specific apoptosis is critical for NAFLD progression. Feldstein and colleagues have found hepatocyte-specific deletion of Bid protects mice from experimental NASH [55]. Animal models for pediatric NAFLD are still not well-established. Recently, Marin et al. [71], used a mouse model for juvenile NAFLD, in which mice were exposed to a HFD with fructose supplemented drinking water from the time of weaning until 16 weeks of age. These mice developed steatosis, hepatic inflammation and fibrosis.

Emricasan and Selonsertib are two anti-apoptotic drugs which have reached Phase II in clinical trial investigations [72]. Emricasan, a pan caspase-inhibitor, improves liver enzyme alanine aminotransferase (ALT) and cytokeratin 18 fragments, a marker of apoptosis. Selonsertib, an inhibitor of apoptosis signal regulating kinase 1 (ASK1), reduces fibrosis, but does not affect ALT levels. Since anti-apoptotic drugs alone only partially suppress or delay disease progression, a need for new therapeutic strategies targeting other modes of cell death are warranted.

Necroptosis, a caspase-independent PCD modality, shares the same initiation route as apoptosis following exposure to death ligands, including TNFα or CD95. However, the subsequent steps of cell death morphologically resemble cellular necrosis [73]. While the direct contribution of caspase-dependent apoptotic cell death to the progression of NAFLD is well established, the distinct contribution of caspase-independent programmed cell death pathways to liver injury and glucose homeostasis are not completely understood. TNFα can initiate either apoptosis or necroptosis, depending on the intracellular energy status, or if caspases are inhibited [74]. Receptor interacting protein kinase 1 (RIP1) and 3 (RIP3) are critical mediators of necroptosis. Both RIP1 and RIP3 possess a Ser/Thr kinase domain which is indispensable for necroptosis and receptor homotypic interaction motif (RHIM); each depends on this mutual interaction. Upon activation, RIP1 and RIP3 undergo a RHIM-dependent interaction followed by RIP3 phosphorylation. Phospho-RIP3 then phosphorylates its downstream effector kinase, mixed lineage kinase domain-like protein (MLKL), which subsequently translocates to the plasma membrane to cause cell death by forming lytic pores (Figure 2) [73]. Recently, RIP3 has been detected in liver biopsies of NASH patients, as well as in the liver of mice exposed to diets deficient in methyl-choline [67], or a high in fat [74], suggesting a role for necroptosis in the pathogenesis of NAFLD/NASH.

The role of RIP3 in mediating hepatocyte injury during NAFLD progression is controversial. Although in vitro studies show that palmitic acid induces RIP3-driven necroptosis in isolated primary hepatocytes [75], results from in vivo studies using RIP3-deficient mice exposed to different feeding models of NAFLD/NASH are contradictory (Table 1). Gautheron et al. [68], first reported that mice deficient in RIP3 are protected from methionine and choline-deficient (MCD) diet-induced hepatocyte injury, hepatic triglyceride accumulation, and fibrosis, characteristics of NASH in human. They have also found that RIP3 drives MCD diet-induced NASH progression through the activation of JNK. In support of the hypothesis that RIP3 contributes to NASH progression, Afonoso et al. [76], also demonstrated that hepatocyte injury and inflammation are ameliorated in RIP3-deficient mice subjected to MCD or high-fat choline-deficient diets (HFCD). Interestingly, none of these murine NASH models cause insulin resistance, a hallmark of NAFLD/NASH in humans. Compared to MCD or HFCD models, feeding mice with a high-fat diet is thought to be a better model to mimic NASH in humans, since mice exposed to this model also develop glucose intolerance in addition to steatosis, inflammation, and fibrosis. Similar to MCD or HFCD diets, HFD feeding also elevated the expression of RIP3 and phospho-MLKL in mouse liver, in parallel to elevations in markers of hepatocyte injury. However, genetic deletion of RIP3 increased hepatic steatosis and ALT/AST in response to HFD, indicating a novel protective role of RIP3 during the progression of NAFLD [75]. The loss of RIP3 also exacerbates HFD-induced glucose intolerance and adipose inflammation in mice. Demonstrating a protective role for RIP3 during diet-induced obesity and insulin resistance, Gautheron and colleagues [77] also found that HFD-induced obesity and adipose inflammation are exacerbated in RIP3-deficient mice. Blocking one modality of cell death can lead to the induction of another cell death pathway. A handful of studies have shown that the inhibition of apoptosis leads to upregulation of the necroptotic machinery. Therefore, upregulation of obesity or glucose intolerance in RIP3-deficient mice could be a consequence of exacerbation in apoptosis, a leading cause of hepatocyte death during NAFLD. Indeed, these studies confirmed that an absence of RIP3 triggered apoptosis in both hepatocytes and adipocytes, resulting in aggravated injury and inflammation. Collectively, these results indicate that RIP3 contributes to diet-induced obesity and insulin resistance, depending on the murine model.

Caspase 1 and NLR family pyrin domain containing 3 (NLRP3) are central mediators of pyroptosis, a highly inflammatory form of programmed cell death [78]. Recently, caspase 4, 5, and 11 have also been implicated. Lipopolysaccharide (LPS)-induced activation of these caspases cleaves gasdermin D (GSDMD), the central executioner of this form of PCD. Cleaved N-terminus of GSDMD translocates to plasma membrane, and forms lytic pores of 12–14 nm inner diameter by interacting with phosphoinositides.

Flavell et al. reported exacerbated liver injury and fibrosis, but attenuated metabolic abnormalities, in mice deficient in Caspase 1 or NLRP3 and exposed to an MCD diet [79]. However, Feldstein and colleagues [80] reported that Caspase 1 deficient mice are protected from liver fibrosis during NAFLD development in response to a Western diet [70]. These results suggest that pyroptosis is diet-specific.

Although pyroptotic cell death is well-established in immune cells, including monocytes, macrophages, and dendritic cells, hepatocyte pyroptosis remains controversial. Making use of genetic manipulation, Wree et al., established that hepatocyte-pyroptosis directly contributes to liver fibrosis. However, within the context of NAFLD or NASH, the role of hepatocyte-specific pyroptosis remains unclear. Future investigations are warranted to clarify mechanisms of hepatocyte pryroptotic cell death.

## 9. NAFLD Treatment: The Role of Gut Microbiome Manipulation

Despite the significant rising epidemic of NAFLD, treatment options with proven efficacy are limited in both adults and children. Weight loss with lifestyle interventions, such as hypocaloric, low-fat, low glycemic index diets and increased physical activity, are the only consistent treatment options shown to reverse the histologic damage caused by NAFLD, including NASH and even fibrosis [34]. Pharmacologic options, such as metformin, vitamin E, omega-3 fatty acids, ursodeoxycholic acid and lipid lowering drugs, have been studied in NAFLD patients with variable results. More importantly, the long-term outcome of these pharmacologic options in patients with NAFLD is not well studied. Due to the lack of patient compliance with lifestyle interventions and variable results with other pharmacologic options, physicians have directed their attention towards other therapeutic options, including the manipulation of gut microbiome using probiotics.

As per The World Health Organization, probiotics are defined as “live microorganisms, [which] when administered in adequate amounts, confer a health benefit on the host” [36]. The multiple mechanisms of action of probiotics in NAFLD have been postulated, including beneficial alteration of gut microbiome, prevention of bacterial translocation by improving intestinal barrier function, and decreasing the production of bacterial products, thereby decreasing overall inflammation [81]. Available probiotics include lactic acid or spore producing bacteria such as Bifidobacteria, Clostridium, or Bacillus gram positive bacteria. These bacterial species are typically packaged into microencapsulated formulations to protect them from gastric acidic or bile damage as they move through the intestinal tract into the colon. Based on the reported probable benefits, multiple animal models have been utilized to investigate the efficacy of probiotics in NAFLD. Among these studies, probiotics are shown to decrease liver/overall inflammation, inflammatory cytokines, insulin resistance, and hepatic steatosis [82,83,84,85]. Promising results from pre-clinical studies led to the evaluation of probiotics in both adults and children with NAFLD. However, randomized controlled studies of these populations are limited. 

Ameta-analysis by Lirussi et al. included two non-randomized trials among adults with NAFLD. This analysis established that the use of VSL#3 or Lactobacilli, plus a prebiotic and vitamin mixture (Bio-Flora) in adults with NAFLD was well tolerated, and improved liver enzymes and markers of lipid peroxidation [86]. Another meta-analysis of four randomized controlled trials which included the use of probiotics in 134 NAFLD/NASH patients showed decreased liver aminotransferases, total-cholesterol, TNF-α, and improved insulin resistance, with no significant changes in BMI [87]. Subsequent to this meta-analysis, Eslamparast et al. conducted a RCT to evaluate the use of synbiotics in 52 adult patients with NAFLD who were randomized to 28 weeks of lifestyle intervention, and either placebo or a synbiotic. Patients in the synbiotic group had decreased aminotransferases, GGT, TNF-α, total nuclear factor κ-B, and fibrosis scores (determined by transient elastography) compared to the placebo group, thereby strengthening the results from previous meta-analyses [88]. In another double-blind RCT, Shavakhi et al. compared the effect of metformin and either a probiotic vs. placebo in 64 adult patients with biopsy-proven NASH. Patients in the metformin-probiotic group had a more significant decrease in aminotransferase, cholesterol, triglyceride levels, and BMI than the metformin-placebo group, thereby indicating a possible augmentation of BMI decreasing potential of metformin when used with probiotics [88].

Similar results are also replicated in pediatric RCTs evaluating the efficacy of probiotics in NAFLD. In an RCT involving 20 children with persistently elevated transaminases and ultrasonography evidence of steatosis, an 8-week treatment with probiotic *Lactobacillus rhamnosus* strain GG (12 billion CFU/day) caused significant improvement in ALT compared to placebo, independent of changes in BMI [89]. In another RCT including 44 children with biopsy-proven NAFLD (22 in VSL# 3 vs. 22 in placebo groups), Alisi et al. demonstrated significant improvement in ultrasound based steatosis after 4 months of treatment with VSL#3^®^, compared to placebo [90]. These results were replicated in another randomized trial involving the use of a probiotic capsule containing a mixture of *Lactobacillus acidophilus*, *Bifidobacterium lactis*, *Bifidobacterium bifidum*, and *Lactobacillus rhamnosus* in obese children with ultrasonographic evidence steatosis. The intervention group showed significant improvement in liver enzymes and lipid profiles [91]. Despite these positive results from the use of probiotics in NAFLD, it is important to understand the caveats of these studies. Available human studies are limited by small sample sizes, heterogeneity in study populations, the lack of randomized controlled trials, and above all, variance in the responses to the studied interventions. Therefore, although probiotics have been proposed as a potential treatment modality in NAFLD, currently there is not enough high quality evidence to back up their utility as a primary treatment for NAFLD [92]. 

Interventions targeted to modulate the bile acid converting bacteria have been investigated in patients to improve serum vitamin D [93], gastrointestinal health [94], and blood cholesterol levels and other symptoms of metabolic syndrome [95]; thus, further investigations into NAFLD are warranted.

There have been a number of studies investigating medicinal plants and natural compounds derived from functional foods (fruits, vegetables) as a means to treat NAFLD and/or its related metabolic conditions [96]. Compounds investigated are known to have anti-inflammatory and antioxidant properties, and thus, could potentially counteract the negative effects of HFD-induced metabolic and organ dysfunction. While many studied compounds have shown benefits in counteracting these negative effects, whether these effects are mediated via alterations in the gut microbiome is not fully known. Red pitaya (*Hylocereus polyrhizus*), rich in the antioxidant betacyanins, was investigated in mice fed a HFD, and compared to mice fed a low-fat diet for 14 weeks [97]. Supplementation reduced weight gain and visceral obesity, and improved hepatic steatosis and insulin resistance. Additionally, modulation of the gut microbiota was noted with a decrease in *Firmicutes*, and increase in *Bacteroidetes* phyla. Shenling Baizhu (SLBZP) powder, a traditional Chinese medicine formula, is a mixture of ten natural medicinal plants which has been found to be clinically efficacious in non-infectious diarrhea and irritable bowel syndrome. Recently, this product was investigated in rats fed an HFD for 16 weeks to determine whether it had any effect on liver injury, inflammation, and the gut microbiome, compared to rats supplemented with a probiotic containing 9 species and/or lipopolysaccharide (LPS) [98]. Supplementation with SLBZP improved body weight, liver enzymes, hepatic fat, cholesterol accumulation, inflammatory markers, and intestinal integrity disrupted by HFD feeding similar to the probiotic supplementation. The relative abundance of the gut microbiota and the short-chain fatty acid producing bacteria were also enhanced with SLBZP supplementation. These data are promising, and further investigations of functional food properties on the modulation of the gut microbiome, and related influences upon metabolic disorders, is warranted. 

## 10. Summary

With a continued rise in obesity and incidence of NAFLD, there remains a great need to develop therapies to mitigate NFALD development and progression. Currently, only diet and lifestyle change have been demonstrated to improve these metrics, but patient compliance is problematic. As more is being discovered about the gut microbiome and its role in obesity and liver disease, future directions in targeting the gut microbiome as a therapeutic option for NAFLD is warranted. Overall, multiple studies show that probiotics might play a role in decreasing liver and overall inflammation in patients with NAFLD. There is no concrete evidence to support probiotic as monotherapy for NAFLD, which has a multi-hit pathophysiology. Nonetheless, manipulation of the gut microbiome using probiotics may be used as combination therapy, in addition to lifestyle interventions and other available natural or pharmacologic options, especially in patients who are struggling with compliance. However, several questions remain unanswered, such as the actual mechanism of action of probiotics in NAFLD, the differences in efficacy between children and adults with NAFLD, the comparison of efficacy of available probiotics, the specific targets of each probiotic, and long-term outcomes. Although promising results along with minimal cost and side effects make probiotics an exciting treatment option for NAFLD, further RCTs with larger sample sizes, long-term follow-up, and assessments of efficacy based on liver histology are needed.

## Figures and Tables

**Figure 1 medsci-06-00047-f001:**
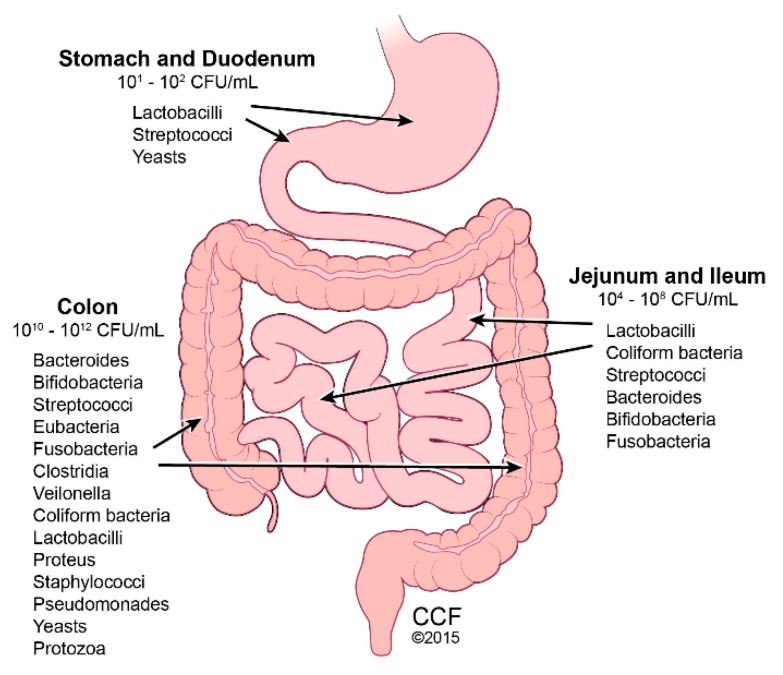
Gut Microbiota Predominance. Reprinted with permission, Cleveland Clinic Center for Medical Art & Photography © 2015–2018. All Rights Reserved. Illustration by David Schumick, BS, CMI.

**Figure 2 medsci-06-00047-f002:**
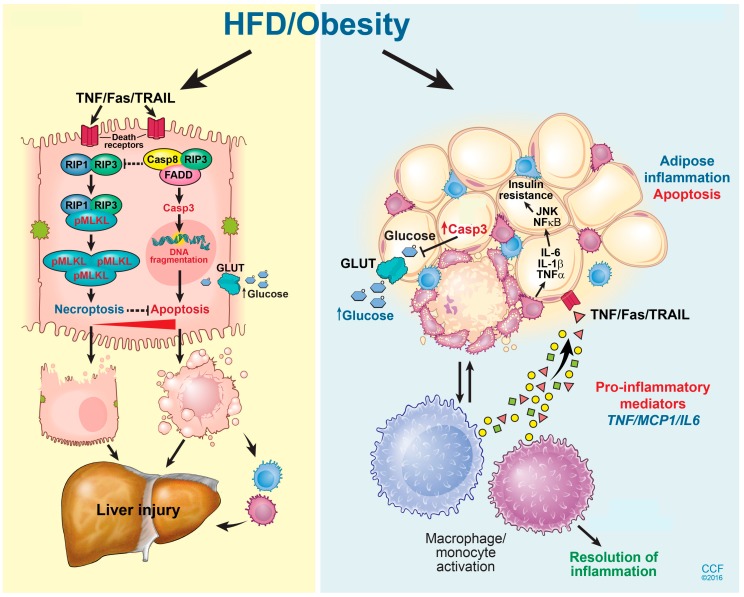
Hepatic cell death pathway and liver injury in response to high fat diet exposure. A HFD induces multiple cell death mechanisms including both caspase-dependent apoptosis and RIP3-driven necroptosis in the liver. Inhibition of RIP3 exacerbates hepatocyte apoptosis following exposure to a HFD, and thus aggravates liver injury and inflammation. A high-fat diet also induces apoptosis in adipocytes. Adipocyte apoptosis drives monocyte/macrophage infiltration, as well as impairs glucose homeostasis. Reprinted with permission, Cleveland Clinic Center for Medical Art & Photography © 2015–2018. All Rights Reserved. Illustration by David Schumick, BS, CMI.

**Table 1 medsci-06-00047-t001:** Differential contribution of RIP3 in NAFLD.

Feeding Models	Outcome	Phenotype	Reference
Western Diet	Protective	Protects against hepatocyte injury, steatosis and fibrosis	[75]
Choline deficient High fat diet (Research Diets; D05010402)	Protective	Suppress adipocyte apoptosis and inflammation	[77]
Methyl choline deficient diet	Deleterious	Induces hepatocyte injury	[68]
Choline-deficient high fat diet	Deleterious	Induces liver injury, inflammation, steatosis and fibrosis	[76]

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
