# Peer review of "The Role of the Gut Microbiome in Nonalcoholic Fatty Liver Disease"

_medsci, 2018, doi:10.3390/medsci6020047_

Round 1

Reviewer 1 Report

This is an actual topic in clinical medicine. However, there are major points, considering these the quality can be improved.

Major points:

1. am missing the topic of obesity including longevity, seen for instance in Okinawa individuals. please check for references, include these and discuss in the text.

2. There is not a single word on CYP 2E1, Please discuss in the text and references pblications.

3. NAFLD is a risk factor of drug induced liver injury, Please check recent references, include these in the text and discuss.

4. There are studies of plants such as Alpinia zerumbet which can be effective in treatment of obesity and NAFLD. please reference an discuss in the text.

5. MEOS may also play a role in NAFLD. Provide references and discuss in the text.

6. How is the intestinal injury explained? in alcoholism the cause is the ingested alcohol.

Author Response

Dear Reviewer:

We greatly appreciate your review and comments.  Please see our responses to your queries below.

Reviewer 1:

Major points:

1.I am missing the topic of obesity including longevity, seen for instance in Okinawa individuals. please check for references, include these and discuss in the text.

Response: The reviewer makes a good suggestion. Accordingly, we included a brief paragraph on obesity in introduction with reference on the Okinawan population as suggested by the reviewer.

2. There is not a single word on CYP 2E1, Please discuss in the text and references publications.

Response: The role of CYP2E1 in NAFLD is now discussed in section 7.

3. NAFLD is a risk factor of drug induced liver injury, Please check recent references, include these in the text and discuss.

Response: We agree with the reviewer that  NAFLD is one of the risk factors for drug-induced liver injury. However, it demands a separate and elaborate discussion in a separate review article. This review is focused on the role of gut-liver axis in NAFLD pathophysiology and therefore we will not include here.

4. There are studies of plants such as Alpinia zerumbet which can be effective in treatment of obesity and NAFLD. Please reference and discuss in the text.

The reviewer makes an excellent point that multiple medicinal plants and functional foods have been investigated in obesity and NAFLD. As this review is focusing on manipulation of the gut microbiome, we have added a section regarding this topic but have limited it to those known to have influenced the microbiome.

Response:

5. MEOS may also play a role in NAFLD. Provide references and discuss in the text.

Response: Thank you for this comment. However, since this paper is solely focused on NAFLD, we have not included the role of microsomal ethanol oxidizing system in development of NAFLD. We believe that it would be more appropriate to discuss it in connection to Alcoholic liver disease (ALD). However, we have discussed here the contribution of CYP2E1, a member of the microsomal enzyme family, in NAFLD pathology.

6. How is the intestinal injury explained? in alcoholism the cause is the ingested alcohol.

Response: Thank you for this comment. We have added information regarding role of HFD induced gut dysbiosis in alterations in intestinal integrity.

Reviewer 2 Report

GENERAL COMMENT

I read with the interest the section on the molecular pathogenesis of NAFLD as related to gut microbiota. However, I found some apparent inconsistencies. For example, many citations and statements refer to pediatric NAFLD, although this is not clarified in the title, which may thus be possibly changed e.g.  “NAFLD in adult and children”.

Moreover, the epidemiology and natural history of NAFLD as well as its relationship with the MetS are incompletely summarized and further citations are suggested to improve the submission under this aspect. Similarly, the introduction fails to accompany the reader into the topic smoothly: suggestions AIMED  to improving THE readability are appended below.

Finally, I would ALSO suggest discussing, in brief, to the role of adipose tissue (and of gut microbiota in diabesity) as a major player in the pathogenesis of NAFLD

SPECIFIC COMMENT

Excessive accumulation of triglycerides in hepatocytes –> these authors should declare how much fat is normal in the normal liver ? (Int J Mol Sci. 2016 Apr 27;17(5). pii: E633. doi: 10.3390/ijms17050633)  and what the biological implications of this accumulation might be (Dig Liver Dis. 2015;47:181-90).

NAFLD is considered a hepatic manifestation of metabolic syndrome -à This as an outdated and partial view of a much more complex and bi-directional phenomenon  (Dig Liver Dis. 2017;49:471-483)

“NAFLD may also be associated with chronic kidney disease, hypothyroidism,  polycystic ovarian syndrome, obstructive sleep apnea (OSA), osteoporosis, and colorectal cancer 6.--> please, note that ref. 6 refers to children. More appropriate citations in adults (in whom most of such complications are indeed observed) are suggested (Metabolism. 2018;79:64-76; World J Gastroenterol. 2017;23:6571-6592;  Eur Respir J. 2017;49(6). pii: 1700546. J Endocrinol Invest. 2015;38:817-25;  Curr Hepatol Rep. 2016;15(2):75-85. Int J Mol Sci. 2016;17(5). Endocrine. 2016;51:211-21).

In this review, we aim to review the recent evidence on gut-liver axis, its role in development and  progression of NAFLD and nutritional approaches to modify the gut microbiome as a strategy to mitigate NAFLD.--> This statement does not fit in with what has been discussed so far. Please, anticipate it by including a paragraph describing what gut microbiota is, how it is studied and what its metabolic impact is deemed to be (Obesity (Silver Spring). 2018 May;26(5):792-800).

The association between obesity and fatty liver has been first reported in 1884 when the presence  of fatty liver was described in a diabetic patient.--> Reference, please ?

NAFLD is believed  to affect approximately one-third of people in the United States with an estimated prevalence of 27- 34%--> This a partial view of a global phenomenon.  A more balanced view of the epidemiology of NAFLD should be provided  ( J Gastroenterol Hepatol. 2018 Jan;33(1):70-85. . J Gastroenterol Hepatol. 2018;33(1):86-98;  Hepatology. 2016;64:73-84

Nat Rev Gastroenterol Hepatol. 2018 ;15:11-20; Dig Liver Dis. 2015;47:997-1006J Gastroenterol Hepatol. 2009;24 Suppl 3:S105-18; )

Keeping up with the rising obesity epidemic, NAFLD has now become the leading cause of chronic liver disease amongst all ages in Westernized countries and has become the second most common cause of liver transplantation among adults à First, based on published data, I am not sure what begets what between NAFLD and obesity (J Gastroenterol Hepatol. 2016;31:936-44; Hepatol Res. 2016 Oct;46(11):1074-1087.). Thus, I would not emphasize obesity alone. Second, Authors may be willing to allude to developing countries as well (Dig Dis Sci. 2015;60(11):3194-202.  Ann Hepatol. 2016;15(5):662-72.  PLoS One. 2017;12(10):e0187033.   Hepatol Int. 2013;7 Suppl 2:755-64. J Gastroenterol Hepatol. 2013;28(1):18-23.   Clin Sci (Lond). 2008 ;115(5):141-50.   World J Gastroenterol. 2006;12(45):7239-49.  Hepatology. 2006 Sep;44(3):521-6. ).

Additionally, gut microbiota is only one among the pathogenic players in NAFLD. Accordingly, a more balanced view is suggested by – at least – alluding to the role of inflamed and expanded adipose tissue in the pathogenesis of NAFLD (Dig Liver Dis. 2017;49:471-483; Gut. 2017;66:1138-1153 J Allergy Clin Immunol. 2013;132:287-94).

Author Response

Dear Reviewer:

We greatly appreciate your comments. Please see below our responses.

Reviewer 2:

GENERAL COMMENT

I read with the interest the section on the molecular pathogenesis of NAFLD as related to gut microbiota. However, I found some apparent inconsistencies. For example, many citations and statements refer to pediatric NAFLD, although this is not clarified in the title, which may thus be possibly changed e.g.  “NAFLD in adult and children”.

Response: The reviewer make a valid point. We included more references from adult literature as mentioned below.

Moreover, the epidemiology and natural history of NAFLD as well as its relationship with the MetS are incompletely summarized and further citations are suggested to improve the submission under this aspect. Similarly, the introduction fails to accompany the reader into the topic smoothly: suggestions AIMED to improving THE readability are appended below.

Response: The reviewer makes excellent suggestions and corresponding changes have been made in the manuscript as mentioned below.

Finally, I would ALSO suggest discussing, in brief, to the role of adipose tissue (and of gut microbiota in diabesity) as a major player in the pathogenesis of NAFLD.

Response: As per reviewers suggestion we have included here a section about the role of adipose tissue in NAFLD pathology in section 6 (page 4, ln 177).

 SPECIFIC COMMENT

Excessive accumulation of triglycerides in hepatocytes –> these authors should declare how much fat is normal in the normal liver ? (Int J Mol Sci. 2016 Apr 27;17(5). pii: E633. doi: 10.3390/ijms17050633)  and what the biological implications of this accumulation might be (Dig Liver Dis. 2015;47:181-90).

 NAFLD is considered a hepatic manifestation of metabolic syndrome - This as an outdated and partial view of a much more complex and bi-directional phenomenon  (Dig Liver Dis. 2017;49:471-483)

Response: The reviewer makes a great suggestion. Therefore, we clarified the percentage of fat in liver required to define NAFLD with a reference. Also, as pointed out, the bi-directional phenomenon of NAFLD and other metabolic diseases is briefly described in introduction with appropriate references.

 “NAFLD may also be associated with chronic kidney disease, hypothyroidism, polycystic ovarian syndrome, obstructive sleep apnea (OSA), osteoporosis, and colorectal cancer 6.--> please, note that ref. 6 refers to children. More appropriate citations in adults (in whom most of such complications are indeed observed) are suggested (Metabolism. 2018;79:64-76; World J Gastroenterol. 2017;23:6571-6592;  Eur Respir J. 2017;49(6). pii: 1700546. J Endocrinol Invest. 2015;38:817-25;  Curr Hepatol Rep. 2016;15(2):75-85. Int J Mol Sci. 2016;17(5). Endocrine. 2016;51:211-21).

Response: As suggested by the reviewer, additional references with citation to adult literature has been added.

 In this review, we aim to review the recent evidence on gut-liver axis, its role in development and progression of NAFLD and nutritional approaches to modify the gut microbiome as a strategy to mitigate NAFLD.--> This statement does not fit in with what has been discussed so far. Please, anticipate it by including a paragraph describing what gut microbiota is, how it is studied and what its metabolic impact is deemed to be (Obesity (Silver Spring). 2018 May;26(5):792-800).

Response:  We do agree with the reviewer’s suggestion. Accordingly, we included a paragraph on obesity as suggested by another reviewer in introduction section and briefly described the relationship of gut microbiome with obesity and NAFLD with relevant references before the above mentioned statement. 

 The association between obesity and fatty liver has been first reported in 1884 when the presence of fatty liver was described in a diabetic patient. --> Reference, please ?

Response: Corresponding reference has been added to the manuscript.

 NAFLD is believed  to affect approximately one-third of people in the United States with an estimated prevalence of 27- 34%--> This a partial view of a global phenomenon.  A more balanced view of the epidemiology of NAFLD should be provided  ( J Gastroenterol Hepatol. 2018 Jan;33(1):70-85. . J Gastroenterol Hepatol. 2018;33(1):86-98;  Hepatology. 2016;64:73-84

Nat Rev Gastroenterol Hepatol. 2018 ;15:11-20; Dig Liver Dis. 2015;47:997-1006J Gastroenterol Hepatol. 2009;24 Suppl 3:S105-18; )

Response: The reviewer makes an excellent point. Therefore, global prevalence of NAFLD in different continents has been included in the epidemiology section with appropriate references.

 Keeping up with the rising obesity epidemic, NAFLD has now become the leading cause of chronic liver disease amongst all ages in Westernized countries and has become the second most common cause of liver transplantation among adults  First, based on published data, I am not sure what begets what between NAFLD and obesity (J Gastroenterol Hepatol. 2016;31:936-44; Hepatol Res. 2016 Oct;46(11):1074-1087.). Thus, I would not emphasize obesity alone. Second, Authors may be willing to allude to developing countries as well (Dig Dis Sci. 2015;60(11):3194-202.  Ann Hepatol. 2016;15(5):662-72.  PLoS One. 2017;12(10):e0187033.   Hepatol Int. 2013;7 Suppl 2:755-64. J Gastroenterol Hepatol. 2013;28(1):18-23.   Clin Sci (Lond). 2008 ;115(5):141-50.   World J Gastroenterol. 2006;12(45):7239-49.  Hepatology. 2006 Sep;44(3):521-6. ).

Response: The reviewer makes a valid point. Therefore, we edited the statement accordingly. Also we discussed about prevalence rates in different countries including a note on prevalence rates in developing nations. However, to give more context to the crux of the article i.e. gut microbiome in NAFLD, we tried to limit our discussion on epidemiology. 

 Additionally, gut microbiota is only one among the pathogenic players in NAFLD. Accordingly, a more balanced view is suggested by – at least – alluding to the role of inflamed and expanded adipose tissue in the pathogenesis of NAFLD (Dig Liver Dis. 2017;49:471-483; Gut. 2017;66:1138-1153 J Allergy Clin Immunol. 2013;132:287-94).

Reviewer 3 Report

To give readers, also those  ouside this specific field, a more balanced view of the topic authors are kindly requested to comment and quote also research in contrast with the optimistically favourable effects of gut flora modifiers as 

evident in recent work, i.e.,Future Microbiol. 2015;10(5):889-902. Systematic review on intervention with prebiotics/probiotics in patients with obesity-related nonalcoholic fatty liver disease.

Author Response

Reviewer 3:

To give readers, also those outside this specific field, a more balanced view of the topic authors are kindly requested to comment and quote also research in contrast with the optimistically favorable effects of gut flora modifiers as evident in recent work, i.e., Future Microbiol. 2015;10(5):889-902. Systematic review on intervention with prebiotics/probiotics in patients with obesity-related nonalcoholic fatty liver disease.

Response: The reviewer makes a reasonable suggestion. So we included details on the limitations of available research on probiotics in the treatment of NAFLD in the manuscript.

Round 2

Reviewer 1 Report

Responses are fairly good, so paper should be accepted now.

Author Response

Thank you for your review.

Best regards.

Reviewer 2 Report

Lines 22-23 The following references should be updated (Anstee, Targher, & Day, 2013; Byrne & Targher, 2015; 22 Miele & Targher, 2015).My suggestion would be: J Hepatol. 2018 Feb;68(2):335-352.

Lines 79-80. (Fazel, Koenig, Sayiner, Goodman, & Younossi, 2016). Further suggested citation Dig Liver Dis.2015 Dec;47(12):997-1006,

Author Response

Thank you for your review.

Lines 22-23 The following references should be updated (Anstee, Targher, & Day, 2013; Byrne & Targher, 2015; 22 Miele & Targher, 2015).My suggestion would be: J Hepatol. 2018 Feb;68(2):335-352.

Response: We appreciate this valuable feedback from the reviewer. Correspondingly, references Anstee, Targher, & Day, 2013; Byrne & Targher, 2015; 22 Miele & Targher, 2015 have been replaced by suggested reference.

Lines 79-80. (Fazel, Koenig, Sayiner, Goodman, & Younossi, 2016). Further suggested citation Dig Liver Dis.2015 Dec;47(12):997-1006,

Response: The reviewer makes a great suggestion. Accordingly, the suggested reference has been added to the manuscript.  

Reviewer 3 Report

Nice job

Author Response

(The authors gave the same response as above.)
